# Brain-Heart Aging During Sleep Predicts Incident Stroke

1st Matteo Saibene*
*Department of Health Technology*
*Technical University of Denmark*
2800 Kgs. Lyngby, Denmark
saima@dtu.dk

2nd Gouthamaan Manimaran*
*Department of Health Technology*
*Technical University of Denmark*
2800 Kgs. Lyngby, Denmark
gouma@dtu.dk

3rd Sadasivan Puthusserypady
*Department of Health Technology*
*Technical University of Denmark*
2800 Kgs. Lyngby, Denmark
sapu@dtu.dk

4th Ying Gu
*Department of Health Technology*
*Technical University of Denmark*
2800 Kgs. Lyngby, Denmark
yingu@dtu.dk

5th Helena Domínguez
*Department of Cardiology*
*Bisperbjerg Hospital*
2100 Copenhagen, Denmark
mdom0002@regionh.dk

6th Martin Ballegaard
*Department of Clinical Medicine*
*University of Copenhagen*
2100 Copenhagen, Denmark
https://orcid.org/0000-0003-3594-1997

7th Jakob E. Bardram
*Department of Health Technology*
*Technical University of Denmark*
2800 Kgs. Lyngby, Denmark
jakba@dtu.dk

*Abstract*—Detecting stroke risk remains a major challenge in preventive medicine. In this work, we introduce a novel computational approach for modeling the effect of aging to identify patients at risk of stroke by analyzing the intricate relationship between brain and heart dynamics during sleep. We analyzed whole-night Polysomnography (PSG) data focusing on sleep stage transitions, to capture changes in cortical and autonomic functions. Using an attention-based model tuned for age estimation, we identify patients at risk of stroke. The model has been tested on 782 patients and a systematic ablation study was performed to evaluate predictive performance across different signal modality configurations and sleep stages.

Results from this study indicate that the patients at risk of stroke show pronounced aging effects, suggesting that Brain-Heart Interaction (BHI) during sleep may be applied on a population level as a novel biomarker to identify patients at risk of stroke.

*Index Terms*—Stroke, Biological Age, Deep Learning

## I. INTRODUCTION

Stroke remains one of the leading causes of death and long-term disability worldwide. It is estimated that over 101 million people are currently living with stroke-related disabilities, and the global economic impact of this condition is staggering, exceeding 721 billion USD [1].

Stroke causes localized brain damage due to cell death from lack of perfusion of oxygenated blood or extravasation of blood [2], [3]. Survivors often face a range of deficits due to damage to eloquent brain regions, including speech, visual impairments, and impaired movements (paresis, spasticity, discoordination) [3], [4]. Given these severe consequences, it is crucial to identify and address risk factors that can lead to a stroke. While current medical tests like blood tests and Electrocardiogram (ECG) can help assess stroke risk, they often focus on cardiovascular health and may miss early, subtle changes in brain function and other systemic factors [5], [6]. This may partially be due to the fact that the stroke may be thrombotic (caused by a blood clot forming in a cerebral artery) or embolic (caused by a clot or debris that forms near the heart and travels to the brain). In the case of a thrombotic stroke, there may be no markers in the ECG of the patient [7], [8]. With advancements in Artificial Intelligence (AI) and data science, new methods are emerging that offer valuable insights into the effects of cardiovascular diseases and stroke [9], [10]. However, most existing early detection methods rely heavily on ECG data, clinical risk factors, and comorbidities, often overlooking processes affecting other organs, such as the brain [11]–[13]. Therefore, exploring an approach that also incorporates brain-derived biomarkers and contextual data to identify early markers of stroke could be highly beneficial.

Sleep and sleep disorders demonstrate significant associations with cardiovascular pathophysiology, including hypertension, autonomic system dysfunction, cardiac arrhythmias, and Congestive Heart Failure (CHF) [14], [15]. In addition, there is growing evidence that sleep disorders are associated with neurodegenerative disorders such as Dementia, Parkinson's

---

* Matteo Saibene and Gouthamaan Manimaran contributed equally to this work.

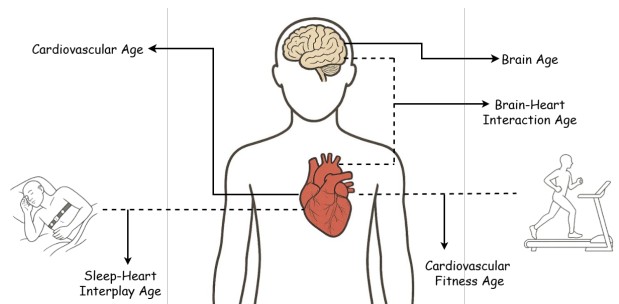

Fig. 1: **Modality-specific aging metrics.** Biological age estimates—e.g., brain age (EEG), heart age (ECG), sleep–heart age (ECG-Sleep Stages), and fitness age—depend on their source data

disease, and Alzheimer's disease [16]–[18]. Polysomnography (PSG) is a systematic procedure that uses Electroencephalography (EEG), ECG and other signal acquisition methodologies to evaluate the underlying causes of sleep disturbances [19]. Sleep stage transitions can be used to characterize physiological markers that reflect the dynamics of the cortical and autonomic nervous systems, including the oscillation of theta ($\theta$) waves, delta ($\delta$) waves, and alpha ($\alpha$) waves and heart rate variations [20]. Therefore, we believe that analyzing the interactions between cortical function and autonomic function, known as Brain-Heart Interaction (BHI), measured with synchronous EEG and ECG signals during sleep, may offer a promising avenue for revealing prognostic markers of stroke.

Previous research on sleep patterns in healthy and sleep apnea subjects, suggests the existence of a bidirectional flow of information from brain to heart [21], [22]. In addition, previous studies have shown that neurological diseases such as Parkinson's, which affect areas of autonomic control, exhibit disturbances in BHIs [23]. Age-related alterations in cardiovascular and neural activity are also associated with an increased risk of mortality and morbidity [24], [25]. However, despite the evidence linking the aging effects of the myocardium with an increased risk of cardiovascular diseases, there is little to no evidence linking BHI to early stroke indicators [9], [26]. *Chronological age* is defined as the number of years since birth and it has been linked to a key factor in clinical risk scores, while *biological age* reflects the individual-specific effects of aging [9]. It is important to recognize that *biological age* estimates depend on the data modality and therefore reflect tissue- or system-specific aging rather than a global measure. For example, age predicted from ECG alone primarily captures cardiovascular aging, whereas multi-modal approaches integrate signals from distinct physiological domains. In a recently published study, Manimaran *et al.* [27] proposed a model to predict biological age by using a combination of ECG and Sleep Stage features (Fig. 1). The model showed associations with cardiovascular diseases, but found no significant link to stroke.

In this work, we hypothesize that the effects of aging on BHI observed during sleep, plays a significant role in stroke

prediction, as sleep is closely linked to cognitive and autonomic processes [28]. Our central hypothesis focuses on sleep stage transitions, which are characterized by profound alteration in both autonomic and cortical systems dynamics [29]. We explore this concept by training a neural network to estimate chronological age using the interplay between EEG and ECG, and treating the estimated age as the biological age during the test phase. We then treat the difference between the chronological age (actual age) and biological age (predicted age) as a potential marker for stroke. By analyzing BHI during sleep stage transitions, we aim to gain new insights into the complex relationship between the cortical and autonomic systems, which could enable early stroke prediction.

In summary, the key contributions of this paper are:

1) We apply an attention-based model that leverages Brain-Heart Interactions (BHIs) to estimate biological age and uses this as a stroke risk factor.
2) We demonstrate that the combined BHIs offers greater predictive value for stroke risk than using either modality independently.
3) We show that BHIs during deep sleep stages yield the strongest predictive power for stroke risk assessment.

## II. MATERIAL AND METHODS

This study was designed as a retrospective analysis of PSG data from the Sleep Heart Health Study (SHHS) dataset [30]. This dataset is part of a longitudinal, multicenter study designed to examine the impact of cardiovascular disorders on sleep and assess their correlation with heightened risks of coronary heart disease, stroke, and all-cause mortality. The dataset includes full night PSG recordings of 5793 subjects during an initial study (SHHS-1) and a follow-up study (SHHS-2).

### A. Data selection

To test our hypothesis, we performed experiments on a subset of the SHHS-1 dataset, focusing only on EEG and ECG signal modalities and subjects without cardiovascular comorbidities at the moment of PSG baseline recording (see Table I). The analysis was limited to PSG recordings that included single-lead ECG data and C3/A2 and C4/A1 EEG signals, sampled at 125 Hz. The Exclusion criteria (EC) for the subset cohort are listed in Table I. We selected cardiovascular-related comorbidities as Exclusion criteria (EC) due to their potential impact on predicting biological age. Based on these criteria (EC1 –to EC6), we identified a subset of 782 subjects without cardiovascular comorbidities at baseline recording denoted as *"healthy at baseline"*.

We stratified participants into *"stroke-free cohort"* (no stroke within 10 years after baseline recording; $N = 674$) and *"incident-stroke cohort"* (stroke within 10 years after baseline recording; $N = 108$) groups for our primary analyses. Using the same 10-year incidence criteria, we also formed Congestive Heart Failure (CHF) cohorts (CHF-free cohort $N = 582$; incident-CHF cohort $N = 200$) and Myocardial Infarction (MI) groups (MI-free cohort $N = 654$; incident-MI

cohort $N = 128$). Outcomes for these cohorts are presented in Section IV.

Training our age-estimation deep learning model was done using the remaining PSG recordings from the SHHS dataset, separate from the test set mentioned above. The data was split into a training and validation cohort (80-20 split) for tuning our deep learning model.

TABLE I: Exclusion criteria for healthy subjects at baseline.

| | |
|---|---|
| **EC1** | Previous incidence of stroke |
| **EC2** | Previous Myocardial Infarction (MI) |
| **EC3** | Any procedures related to heart attack prior to baseline |
| **EC4** | Previous Revascularization procedures prior to baseline |
| **EC5** | Any Congestive Heart Failure (CHF) episodes prior to baseline |
| **EC6** | Any number of Angina Episodes prior to baseline |

### B. Preprocessing

The PSG recordings of all subjects in the *"healthy at baseline"* group were preprocessed. Both EEG and ECG signals underwent filtering using a zero-phase notch filter at 50 Hz to mitigate powerline interference. Subsequently, the signals were filtered with a bandpass zero-phase filter employing a Hamming window, characterized by a maximum passband ripple of 1 dB, a minimum stopband attenuation of 60 dB, and a transition bandwidth of 0.50 Hz with 6 dB attenuation in the transition band. For the EEG signals, a passband of 1 to 45 Hz was applied, while a passband of 4 to 45 Hz was used for the ECG signals. After filtering, 30-second segments corresponding to each sleep stage transition were extracted for every subject as shown in Fig. 2. These segments encompassed the 15-second intervals before and after transitions between different sleep stages, such as from a Sleep Stage 3 to 4. Given that individual subjects exhibited multiple stage transitions throughout their sleep study, each unique transition was systematically saved as a separate file for subsequent analysis.

### C. Model description

Our method focuses on identifying the correlation between the input ECG and EEG signals to derive a feature vector for age prediction. Previous studies have achieved this using convolutional methods on N-lead ECG signals, processing each lead separately and then combining the information from all leads at the end using attention mechanisms [31]. In contrast, our approach seeks to establish correlations at every stage of the encoder. To achieve this, we employ the Local-lead Attention model [32], which allows for a more integrated and continuous correlation analysis throughout the encoding process.

We adapted the local-lead attention transformer that we recently introduced for 12-lead ECG arrhythmia classification to the multi-modal setting of overnight PSG. The model treats the physiological channels—two EEG channels, a single-lead ECG (lead II)—as an independent "lead" that shares a common processing backbone while retaining lead-specific parameters.

A depth-wise separable 1-D convolution (kernel $= 16$, stride $= 8$) embeds each lead into $D = 64$-dimensional patches, yielding $T' = 480$ tokens per lead. Instead of global self-attention, we apply *local* self-attention windows of width $W = 19$ tokens with a stride $S = 2$ (depending on the depth of the model), reducing complexity from $\mathcal{O}(T'^2)$ to $\mathcal{O}(W^2 T'/S)$. Within every window, each query attends not only to keys from the same lead but also to temporally co-located keys from all other leads as shown in the red-highlighted region of the signals in Fig. 2. This "cross-lead" mechanism enables the model to capture cardio-cortical coupling without inflating the parameter count.

*Local-lead attention*: Given an input $Z^0 \in \mathbb{R}^{T \times D}$, where $T$ represents the time length and $D$ the feature vector dimension, $Z^0$ is projected using matrices $W_Q \in \mathbb{R}^{D \times D_q}$, $W_K \in \mathbb{R}^{D \times D_k}$, and $W_V \in \mathbb{R}^{D \times D_v}$. These projections yield the feature representations $Q$, $K$, and $V$, corresponding to the query, key, and value, respectively, with $D_k = D_q$. The outputs $Q$, $K$, and $V$ are computed as follows:

$$\mathbf{Q} = \mathbf{Z}^0 \mathbf{W}_Q, \quad \mathbf{K} = \mathbf{Z}^0 \mathbf{W}_K, \quad \mathbf{V} = \mathbf{Z}^0 \mathbf{W}_V, \quad (1)$$

and local attention is computed as;

$$\mathbf{S} = \mathrm{softmax}\left(\frac{\mathbf{Q}\mathbf{K}^T}{\sqrt{D_q}}\right) \mathbf{V}. \quad (2)$$

This local lead-attention is extended to all signals in the input $\{S^1, S^2, \ldots\}$, each having a multi-scale representation $\{Z^1, Z^2, \ldots\}$, with the same window size across scales. As features are downsampled, a fixed window size corresponds to an increasingly broad temporal range, enabling efficient hierarchical aggregation of local patterns across multiple signals.

Hidden states from all leads are concatenated, pooled by a global multi-head attention layer (4 heads), and fed to a linear regression head that outputs BHI-age.

### D. Implementation

All analyses were done with Python v.3.9.18. The model is developed using PyTorch and has been trained for 50 epochs on an NVIDIA Tesla V100. The batch size is set to 128 with a learning rate of 0.0001. For the task of age estimation, we scale the outputs to $0 - 1$ values and use the Mean Squared Error (MSE) loss as follows:

$$\mathrm{MSE}(y, \hat{y}) = \frac{1}{N} \sum_{i=1}^{N} (y_i - \hat{y}_i)^2, \quad (3)$$

where $N$ is the number of data points in the batch, $y$ is the actual chronological age, and $\hat{y}$ the predicted biological age.

### III. RESULTS

We conducted an evaluation of stroke risk stratification using three distinct approaches: (i) ECG alone, (ii) EEG alone, and (iii) a combination of ECG and EEG. Following the methodology outlined in Section II, we calculated

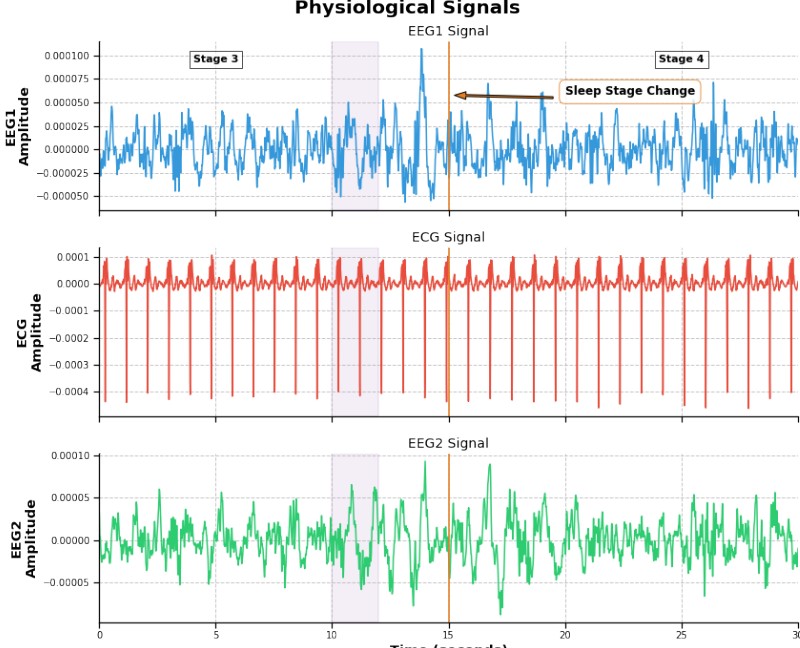

Fig. 2: **Representative 30-s epoch of PSG signals around a sleep-stage transition.** Top and bottom panels show two EEG derivations, middle panel shows ECG lead II. The shaded region denotes the local attention window, and the vertical line marks the annotated transition from stage 3 to stage 4.

TABLE II: Comparison of combined EEG and ECG, ECG-only, and EEG-only measures at different time windows before stroke. The lowest $p$-values per time window for each test are highlighted by cell shading.

| Time (yrs) | Measure | Mean Difference $\pm$ SD | $p$-value |
|---|---|---|---|
| 3.0 | EEG and ECG | $3.03 \pm 4.67$ | 0.005 |
|  | ECG | $2.83 \pm 4.98$ | 0.013 |
|  | EEG | $2.26 \pm 4.35$ | 0.015 |
| 5.0 | EEG and ECG | $1.89 \pm 4.66$ | 0.015 |
|  | ECG | $1.59 \pm 4.79$ | 0.053 |
|  | EEG | $1.25 \pm 4.28$ | 0.096 |
| 10.0 | EEG and ECG | $0.93 \pm 4.47$ | 0.089 |
|  | ECG | $0.72 \pm 4.55$ | 0.237 |
|  | EEG | $0.52 \pm 4.05$ | 0.276 |

the difference between model-predicted biological age and chronological age, and analyzed its correlation with stroke incidence over various time intervals preceding the event. Statistical analysis was conducted by first assessing the normality assumption using the Shapiro–Wilk test ($\alpha = 0.01$) [33]. Upon identifying a deviation from normality, group comparisons were conducted using the non-parametric Mann–Whitney U test [34] and adjusted for multiple comparisons using the False Discovery Rate (FDR) method.

As demonstrated in Table II, while single-modality approaches (ECG or EEG alone) exhibit some predictive capability, the combined ECG and EEG measure consistently provides superior discrimination. Notably, at a 3-year interval,

the combined ECG and EEG approach shows highly significant differences between future stroke patients and controls ($p = 5 \cdot 10^{-3}$), outperforming single-modality estimates.

The difference between chronological and estimated biological age, treated as a continuous variable and adjusted for age and gender, is associated with a Hazard Ratio (HR) of 1.25 (95% CI: $1.09 - 1.44$, $p < 0.005$ ) as shown in Table. III. This indicates that for every one-point increase in estimated age relative to chronological age, the risk of stroke increases by 24%. Additionally, there is no significance for ECG ($p = 0.07$) and EEG ($p = 0.12$) separately.

We further adjusted BHI-age for the individual covariates comprising the CHADS-VASc score [35], [36] in Table. IV. Even after controlling for these established stroke risk factors, BHI-age remained significantly associated with incident stroke.

TABLE III: Cox proportional-hazards models adjusted for chronological age, sex, race show that each 1-year increment in the BHI-age difference derived from the combined EEG + ECG signal confers the highest risk, whereas single-modality estimates are weaker and non-significant.)

| Measure | Hazard Ratio (HR) | 95% CI | $p$-value |
|---|---|---|---|
| ECG+EEG | **1.25** | **[1.09, 1.44]** | **<0.005** |
| ECG | 1.13 | [0.99, 1.28] | 0.07 |
| EEG | 1.17 | [0.96, 1.42] | 0.12 |

TABLE IV: Multivariable Cox regression of incident stroke with clinical covariates. Among all covariates—including age, sex, AHI<15, hypertension, atrial fibrillation, blood pressure, diabetes, smoking, LVH and Chronic Obstructive Pulmonary Disease (COPD)—only BHI-age significantly better predicts stroke risk.

| Variable | HR | 95% CI | *p*-value |
|---|---|---|---|
| Brain-Heart Interaction Age | 1.26 | [1.09, 1.45] | <0.005 |
| Age | 0.85 | [0.73, 0.98] | 0.03 |
| Gender | 0.78 | [0.48, 1.28] | 0.33 |
| Race | 1.36 | [0.82, 2.26] | 0.24 |
| AHI<15 | 1.46 | [0.86, 2.46] | 0.16 |
| Hypertension Medication | 0.97 | [0.62, 1.53] | 0.91 |
| Atrial Fibrillation | 0.67 | [0.36, 1.23] | 0.20 |
| Systolic Blood Pressure | 1.00 | [0.99, 1.01] | 0.79 |
| Diabetes | 0.86 | [0.51, 1.47] | 0.59 |
| Cigarette smoking | 0.99 | [0.99, 1.00] | 0.37 |
| Left Ventricular Hypertrophy | 1.55 | [0.81, 2.97] | 0.19 |
| COPD | 0.88 | [0.75, 1.04] | 0.14 |

TABLE V: Ten-Year Pre-Stroke Results by Sleep Stage Using Combined EEG and ECG Measures

| Sleep Stage | Mean Difference | *p*-value |
|---|---|---|
| REM | 0.89 | 0.171 |
| Stage 1 | 0.75 | 0.201 |
| Stage 2 | 0.92 | 0.085 |
| Stage 3 | 1.05 | 0.081 |
| Stage 4 | 2.39 | 0.009 |

Figure 3 presents survival curves stratified into two groups: one where the estimated age exceeds the chronological age (indicating the patient appears older than their actual age) and one where it does not. These survival curves were derived using Cox proportional hazards models, formulated as $S_0^{KM}(t)^{z(x)}$, where $S_0^{KM}(t)$ represents the baseline survival function, and $z(x) = \exp(\beta_0 + \beta_1 x_1 + \cdots + \beta_n x_n)$ captures the hazard ratio as a function of covariates. Survival functions were plotted by stratifying groups based on whether the estimated age exceeded or did not exceed the chronological age, with the estimated hazard ratios corresponding to the relative risk in each group. This survival analysis demonstrates a significantly higher risk in the group where the estimated age is greater than the chronological age, underscoring the association between age discrepancy and stroke risk.

We further explored whether specific sleep stages provide stronger discriminative power between stroke-free and incident-stroke groups. As shown in Table V, stratifying data by sleep stage at the ten-year window reveals robust significance across all examined stages. Deep sleep stage (stage 4) exhibit particularly pronounced significant differences ($p < 0.05$) between these groups. These findings suggest that integrating EEG and ECG signals not only enhances early stroke risk prediction but that certain sleep stages may offer especially sensitive physiological markers for this task.

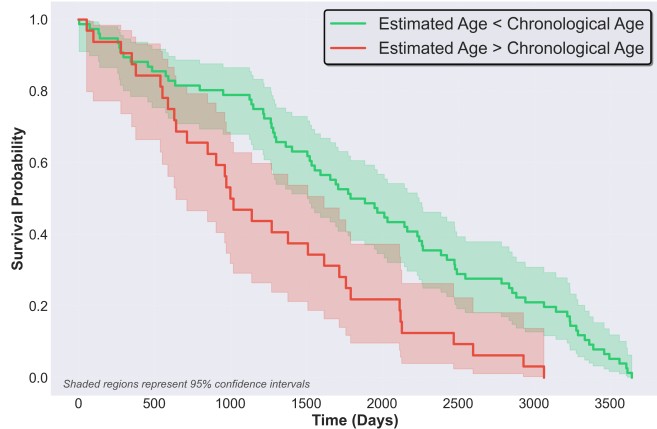

Fig. 3: Survival curves comparing groups where estimated age exceeds chronological age versus where it does not.

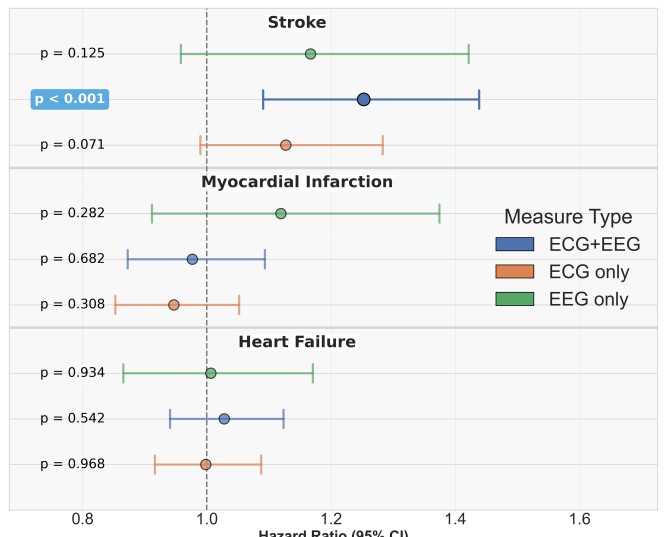

Fig. 4: Hazard ratios (95% CI) per one-year increase in biological age for 10-year risk of stroke, MI, and CHF. Only BHI-age predicts stroke ($p < 0.001$); no measure predicts MI or CHF. The dashed line marks Hazard Ratio (HR) = 1.

## IV. DISCUSSION

In this study, we hypothesized that the aging effects observed in BHI through PSG recordings could serve as a marker for stroke risk. As illustrated in Table II, a significant difference in predicted biological age was observed between the stroke-free and incident-stroke subgroups within the three years preceding stroke onset. The local lead-attention model employed in this study demonstrated that combining EEG and ECG provided significantly better predictive power compared to using EEG or ECG alone. This finding suggests that BHI measured with both EEG and ECG has a significantly higher potential for predicting aging effects in the stroke population.

However, as the time from baseline to stroke onset increases (beyond three years), the predicted age difference diminishes. Notably, the age difference between the stroke-

free and incident-stroke subgroups becomes non-significant after ten years from the baseline recording. Consequently, risk stratification becomes progressively more challenging for strokes occurring further from the baseline recording.

Additionally, we analyzed the data to determine which sleep stage might be more relevant for assessing stroke risk. At the three-year mark, no significant differences were observed across sleep stages, indicating that BHI does not change significantly across different sleep stages and that all sleep stages might be used to capture stroke risk. However in the 10-year mark, Stage 4 is the only sleep stage with significant p-values with a larger mean difference in predicted age as compared to other sleep stages. Although, this may be attributed to the fact that many subjects in the study did not experience transitions into Stage 4.

To assess whether BHI-age has prognostic value beyond stroke, we fitted separate Cox proportional-hazards models for 10-year incidence of MI, CHF, and stroke, each adjusted for age, sex, and other key demographic covariates. As shown in Figure 4, elevated BHI-age was a significant predictor of stroke risk but did not predict MI or CHF. Moreover, when we estimated biological age using ECG-only or EEG-only models, neither modality alone showed significant associations with any of the three cardiovascular outcomes. These findings underscore that combined EEG-ECG coupling—our BHI measure—offers uniquely strong prognostic power for stroke compared to other major cardiovascular diseases.

These findings underscore the importance of integrating EEG and ECG signals for early stroke risk prediction and suggest that certain sleep stages, particularly Stage 4, may offer especially sensitive physiological markers for this purpose.

### A. Limitations and Future Improvements

In this study, we defined the stroke-free subgroup solely based on the absence of heart-related diseases. Consequently, our analysis does not account for other confounding factors and comorbidities that may influence patient outcomes, such as kidney disease and others. Future research should incorporate alternative sleep datasets that provide detailed information on comorbidities, enabling the definition of a more representative stroke-free subgroup.

Despite these limitations, our study yielded meaningful results in stroke risk up to five years from baseline PSG recording. Unlike other sleep models that extract features throughout the entire night of PSG, our approach utilizes $30s$ transition segments between each sleep stage. This method substantially reduces the amount of data the model processes, allowing for more efficient and faster training.

In future work, we aim to validate the results and models of this study using additional datasets as they become available. This will help ensure the robustness and generalizability of our findings, ultimately contributing to more accurate stroke risk prediction and better patient outcomes.

## V. CONCLUSIONS

In this study, we introduced a novel approach to assessing stroke risk. By leveraging an attention-based model that utilizes BHI to predict the biological age of the brain and heart, we estimated aging effects based on the age prediction difference between stroke-free and incident-stroke subgroups. Our approach revealed a significant age difference between these populations within three years after the PSG recording, suggesting that stroke risk can be detected within this time-frame. Unlike previous studies that use entire sleep recordings for age estimation [37], [38], our method focuses on transitions between different sleep stages and BHI. This not only reduces the overall training process but also highlights BHI as a potential biomarker for stroke. Our findings demonstrate that understanding BHI and the intricate interplay between cortical and autonomic functions opens new avenues for significant progress in the diagnosis of stroke. This work underscores the importance of integrating advanced modeling techniques with physiological data to enhance early stroke risk prediction and improve patient outcomes.

## VI. ACKNOWLEDGMENT

This research has been funded by Region Zealand Denmark as part of the 'AI in Neurology' project, and the Innovation Fund Denmark as part of the CATCH project (Project No. #1061-00046B) and the Copenhagen Center for Health Technology (CACHET).

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
