# OpenReview forum: "Brain–Heart Aging During Sleep Predicts Incident Stroke"
_IEEE.org/EMBS/BHI/2025/Conference — BHI 2025_

### Official Review · Reviewer_1QkT · 2025-07-10
**Brain–Heart Aging During Sleep Predicts Incident Stroke**

**Confidence:** 4
**Clarity Of Writing:** great
**Clinical Significance:** good
**Methodological Novelty:** great
**Overall Rating:** 8

**Experiments And Results:**

good

**Questions For The Authors:**

- what do you think about other potential factors beyond cardiovascular health? (neurological, kidney disease)
- why do you think sleep stage 4 yielded strongest response?

**Strengths:**

- the study introduces a new biomarker oupling EEG and ECG during sleep, that reportedly can predict the risk of stroke 3-5 years before the event.
- using multi-modal and transformer based models as the most recent approaches makes it a very interesting case for audience.
- the possibility of using short-term PSG data instead of a full-night collection makes it more accessible to larger populations.

**Summary Of The Paper:**

current stroke risk prediction tools focus on cardiovascular indicators and usually miss early neurological markers. This paper proposes leveraging attention-based methods to model biological age from EEG-ECG during sleep, to determine if aging effects are captured via these modalities. The proposed approach basically uses the differene between predicted biological age and the actual age, as an estimate for risk score.
The study utilizes PSG data from 782 healthy individuals from the SHHS dataset and trained a model to capture brain-heart interactions, showing a 25% increase in stroke risk for every 1 year increase in predicted BHI-age. The authors reported sleep stage-4 as the most sensitive window for their stroke-risk estimator.

**Weaknesses:**

- Potential bias seems to be the case, as "healthy at baseline" group was only defined by cardiovascular comorbidities, ignoring the potential effect of other diseases cuash as neurological or kidney diseases.
- while the attention model is powerful, the physiological interprettation of the learned features remained limiter and no discussion awas provided regarding integration into clinical workflow.
- Focusing on the predictive power of the model, authors have not provided enough insights on the mechanistics of how and why brain-heart coupling mediates stroke risk.

---

### Official Review · Reviewer_51iZ · 2025-07-14
**The manuscript introduces an attention-based EEG + ECG model that derives a sleep-specific Brain–Heart Interaction age (BHI-age). In a retrospective analysis of 782 “cardiovascular-healthy” participants from the Sleep Heart Health Study (SHHS), the excess of BHI-age over chronological age was associated with a 24 % increase in 10-year stroke risk per additional year of divergence, outperforming single-modality EEG- or ECG-based ages. The idea of leveraging multi-modal ageing signatures during sleep for stroke prediction is timely and clinically relevant, but the work would benefit from clearer methodological description, external validation, and deeper comparison to established risk scores.**

**Confidence:** 5
**Clarity Of Writing:** great
**Clinical Significance:** good
**Methodological Novelty:** good
**Overall Rating:** 7
**Final Rating:** 8

**Experiments And Results:**

great

**Questions For The Authors:**

1. Can the trained model be evaluated on another PSG cohort (e.g., MESA Sleep)? Demonstrating reproducibility would markedly improve confidence and could raise the overall rating.
2. Given that many subjects never reach Stage N3/N4, how is class imbalance managed, and does this bias hazard-ratio estimates?
3. Could you supply attention or saliency maps highlighting EEG–ECG features that drive high BHI-age? Concrete examples would strengthen methodological novelty and reader trust.
4. Beyond AHI, were periodic limb movements or insomnia symptoms considered? Clarifying would help gauge robustness.

**Strengths:**

1. Combining EEG and ECG to derive a tissue-integrated ageing metric during sleep is original and grounded in emerging literature on biological age from cardiovascular and neural signals.
2. Prior studies link sleep disorders, autonomic dysregulation, and cerebrovascular outcomes, supporting the focus on stage transitions.
3. Limiting input to transition segments reduces computational load compared to whole-night modelling while retaining salient cardio-cortical coupling.
4. Separate EEG-only and ECG-only models, plus stage-specific analyses, clarify each modality’s contribution.
5. Normality testing, non-parametric comparisons, and multivariable Cox models are appropriate for the endpoints.

**Summary Of The Paper:**

The authors preprocess overnight polysomnography by extracting 30 s windows around every sleep-stage transition and feed two EEG derivations and a lead-II ECG into a local-lead attention transformer. The network is trained to estimate chronological age on the remaining SHHS recordings; at test time, the difference between predicted and true age is used as a biological-age delta (BHI-age). In 674 stroke-free and 108 incident-stroke subjects followed for up to ten years, larger BHI-age deltas (especially during deep-sleep stages 3–4) were linked to future stroke, yielding a hazard ratio of 1.25 (95 % CI 1.09-1.44, p ≪ 0.005). No association was seen for myocardial infarction or congestive heart failure.

**Weaknesses:**

1. All results come from a single cohort; without validation on independent datasets such as MESA Sleep the generalisability is uncertain.
2. Participants were screened only for overt cardiovascular disease, leaving other stroke-related conditions (e.g., chronic kidney disease, COPD) as possible confounders.
3. Stroke sub-types are not separated, missing a chance to test whether haemorrhagic and ischaemic events share the same biomarker behaviour.
4. No interpretability visualisations (e.g., attention maps) are provided, despite growing consensus that explanation is essential for transformer-based EEG models.
5. Obstructive sleep apnoea severity is a well-known stroke risk factor but is only superficially adjusted for, leaving residual confounding likely.

---

### Official Review · Reviewer_7yrx · 2025-07-17
**Promising brain-heart interaction approach with significant methodological limitations requiring major revision**

**Confidence:** 4
**Clarity Of Writing:** good
**Clinical Significance:** good
**Methodological Novelty:** good
**Overall Rating:** 5
**Final Rating:** 7

**Experiments And Results:**

fair

**Questions For The Authors:**

Can you provide evidence that 30-second sleep transition segments contain more stroke-predictive information than stable sleep periods? What is the biological rationale for excluding stable sleep data?

How do you address the multiple comparisons across time windows and sleep stages? Would your key findings (p=0.0005 at 3 years) remain significant after appropriate correction?

Do you have plans to validate these findings on independent sleep study datasets? How would you address potential overfitting to SHHS population characteristics?

Can you provide feature importance analysis or attention maps showing which specific EEG/ECG patterns drive the age predictions? Why is the model stroke-specific but not predictive of other cardiovascular events?

Time-dependent degradation: How do you explain the diminishing predictive power over time? Does this limit clinical applicability for long-term stroke prevention?

**Strengths:**

The focus on brain-heart interactions during sleep stage transitions provides a biologically plausible mechanism linking autonomic-cortical coupling to stroke risk, which is conceptually innovative.

Use of the established SHHS dataset with careful exclusion of cardiovascular comorbidities at baseline creates a well-defined "healthy at baseline" cohort for testing the hypothesis.

The systematic comparison of EEG-only, ECG-only, and combined EEG+ECG approaches demonstrates the added value of multimodal brain-heart interaction modeling.

The approach of analyzing 30-second transition segments rather than entire night recordings offers computational advantages while potentially capturing key physiological dynamics.

Use of non-parametric tests following normality assessment and Cox proportional hazards modeling with covariate adjustment follows appropriate statistical practices.

**Summary Of The Paper:**

This paper proposes using brain-heart interactions (BHI) during sleep stage transitions to predict stroke risk through biological age estimation. The authors analyzed polysomnography data from 782 subjects in the Sleep Heart Health Study, focusing on 30-second segments around sleep stage transitions. They employed a local-lead attention transformer to process combined EEG and ECG signals for age estimation, with the difference between predicted biological age and chronological age serving as a stroke risk biomarker. The study found that patients with incident stroke within 10 years showed higher biological age estimates than stroke-free controls, with a hazard ratio of 1.25 (p<0.005). The effect was strongest for combined EEG+ECG compared to single modalities and appeared most pronounced during deep sleep stages.

**Weaknesses:**

Restricting analysis to only 30-second sleep transition segments may discard valuable information from stable sleep periods. No evidence is provided that transitions are more informative than stable sleep states, representing a fundamental design limitation.

All analyses use variants of the same SHHS dataset. No external validation on independent sleep study populations is provided, severely limiting generalizability claims.

No analysis of which specific EEG/ECG features drive the age predictions or why the model is stroke-specific is provided.

---

### Official Review · Reviewer_L3Tb · 2025-07-22
**Brain–Heart Aging During Sleep Predicts Incident Stroke**

**Confidence:** 5
**Clarity Of Writing:** great
**Clinical Significance:** great
**Methodological Novelty:** excellent
**Overall Rating:** 7

**Experiments And Results:**

great

**Questions For The Authors:**

no questions

**Strengths:**

The manuscript describes aging effects based on the age prediction difference between stroke-free and incident-stroke subgroups.
It has been found that there is a significant age difference between these populations within three years after the PSG recording,
suggesting that stroke risk can be detected within this timeframe.
Good paper suggests that BrainHeart Interaction (BHI) during sleep may be applied on a opulation level as a novel biomarker to identify patients at risk of stroke.

**Summary Of The Paper:**

The manuscript describes aging effects based on the age prediction difference between stroke-free and incident-stroke subgroups.
It has been found that there is a significant age difference between these populations within three years after the PSG recording,
suggesting that stroke risk can be detected within this timeframe.
Good paper suggests that BrainHeart Interaction (BHI) during sleep may be applied on a opulation level as a novel biomarker to identify patients at risk of stroke.

**Weaknesses:**

no weaknesses